# Proposals on Kaplan–Meier plots in medical research and a survey of stakeholder views: KMunicate

Tim P Morris ![ORCID],[1] Christopher I Jarvis ![ORCID],[2] William Cragg ![ORCID],[3] Patrick P J Phillips ![ORCID],[4] Babak Choodari-Oskooei ![ORCID],[1] Matthew R Sydes ![ORCID] [1]

[1]MRC Clinical Trials Unit at UCL, UCL, London, UK
[2]Department of Infectious Disease Epidemiology, London School of Hygiene and Tropical Medicine, London, UK
[3]Clinical Trials Research Unit, University of Leeds, Leeds, UK
[4]Center for TB, University of California San Francisco, San Francisco, California, USA

**Correspondence to**
Dr Tim P Morris;
tim.morris@ucl.ac.uk

## ABSTRACT

**Objectives** To examine reactions to the proposed improvements to standard Kaplan–Meier plots, the standard way to present time-to-event data, and to understand which (if any) facilitated better depiction of (1) the state of patients over time, and (2) uncertainty over time in the estimates of survival.

**Design** A survey of stakeholders' opinions on the proposals.

**Setting** A web-based survey, open to international participation, for those with an interest in visualisation of time-to-event data.

**Participants** 1174 people participated in the survey over a 6-week period. Participation was global (although primarily Europe and North America) and represented a wide range of researchers (primarily statisticians and clinicians).

**Main outcome measures** Two outcome measures were of principal importance: (1) participants' opinions of each proposal compared with a 'standard' Kaplan–Meier plot; and (2) participants' overall ranking of the proposals (including the standard).

**Results** Most proposals were more popular than the standard Kaplan–Meier plot. The most popular proposals in the two categories, respectively, were an extended table beneath the plot depicting the numbers at risk, censored and having experienced an event at periodic timepoints, and CIs around each Kaplan–Meier curve.

**Conclusions** This study produced a high response number, reflecting the importance of graphics for time-to-event data. Those producing and publishing Kaplan–Meier plots—both authors and journals—should, as a starting point, consider using the combination of the two favoured proposals.

## INTRODUCTION

Kaplan–Meier (KM) plots are ubiquitous in medical research, depicting the estimated cumulative proportion of people surviving over time.[1] [2] This is sometimes presented overall, but frequently within groups, such as randomised arms of a clinical trial. For a clear and simple description of how the KM estimate is calculated, see Ref. 3. In producing even a simple KM plot, there are many choices to be made, leading to wide variation in presentation quality.

### Strengths and limitations of this study

► This study made several proposals to improve the information conveyed by Kaplan–Meier plots for survival data. Unlike many proposals for graphics, the study involved a survey of stakeholders' opinions.

► A total of 1174 people participated in the survey representing diverse professions, geographical locations and amounts of experience.

► As a web-based survey for which participants selected themselves, it is not possible to know the number that might have participated and therefore the response proportion for this survey is unknown.

Figure 1 gives one example of a KM plot (based on data from the RT01 trial).[4] Box 1 outlines the basic anatomy of a KM plot and highlights some of the choices to be made for readers who are unfamiliar.

The utility of a KM plot depends on who is using it and their purpose. Potential users may be: members of data monitoring committees considering interim data; systematic reviewers extracting data for meta-analysis; trial designers looking for information from relevant patients for sample size calculations; clinicians trying to understand and communicate survival to their patients; and those interested in the value of an estimand that was not reported, such as restricted mean survival times. Even when produced with care, key information may still be lacking for certain readers. It may seem that KM plots do not require much 'learning to read'. However, we have many times been asked by collaborators how to read them.

We can learn a lot from a KM plot: the estimated survival fraction at various times; the difference in survival fractions between two groups; quantiles of survival time; suggestions that hazard functions may be non-proportional.[5] It is even possible to reconstitute (data similar to) the underlying survival data based on KM plots, often very accurately.[6]

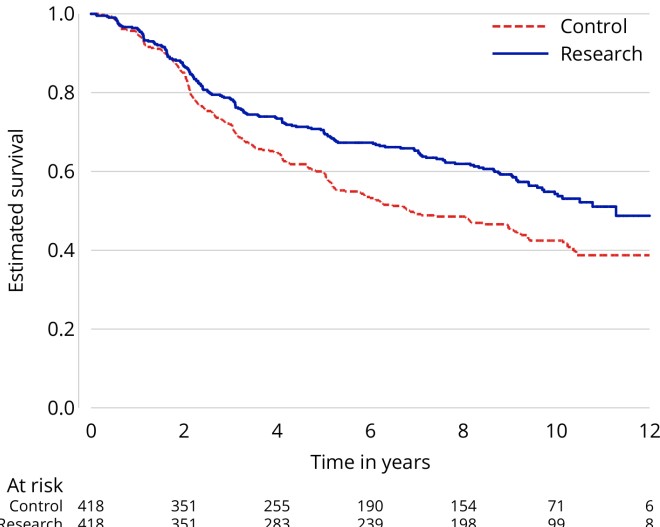

**At risk**

| | | | | | | | |
|---|---|---|---|---|---|---|---|
| Control | 418 | 351 | 255 | 190 | 154 | 71 | 6 |
| Research | 418 | 351 | 283 | 239 | 198 | 99 | 8 |

**Figure 1** An example of a Kaplan–Meier plot from the RT01 trial.

Despite the above strengths, there are many issues which may hinder interpretation of KM plots. The two key factors, and the focus of this work, are to communicate clearly:

1. The number of participants at risk, censored and having experienced an event at specific times, over time.
2. The uncertainty of the KM estimate over time.

These aspects could aid interpretation and increase the amount of information conveyed by KM plots. We believe a central problem is that a standard KM plot does not clearly show that the right-hand portion of the curve (at later time points when there are usually considerably fewer patients at risk) is estimated with much greater uncertainty than the left-hand portion of the curve. As a consequence, we are concerned that many consumers of KM curves place undue emphasis on differences between curves at these later time points when differences are much more likely due to chance.

As a snapshot describing recent practice, we reviewed the KM plots presented in articles published in the BMJ, JAMA (The Journal of the American Medical Association), The Lancet and NEJM (The New England

**Box 1 Anatomy of a Kaplan–Meier plot**

In figure 1, the vertical axis runs from 0 to 1 and the horizontal from 0 to 12 years post randomisation (though this was not the longest follow-up available). The Kaplan–Meier estimate for the control arm is depicted by a red-dashed line and for the research arm by a solid blue line. The 'curves' are stepped over time because the estimate changes only at times when an event has occurred. These steps become more pronounced over time as more participants are censored. Beneath the horizontal axis is a table that reports the number of participants still 'at-risk' at specific time points (here 0, 2, 4, 6, 8, 10 and 12 years), that is, they are still in follow-up at this time point, not having had an event or been censored. In figure 1, after 10 years, there remain 71 and 99 participants at risk of an event in the control and research arms, respectively.

Journal of Medicine) during 2013. In total, there were 50 randomised, superiority trials with a time-to-event primary outcome. The two dominant specialities were cardiovascular disease (22 trials, 44%) and cancer (11 trials, 22%). Forty-seven plots (94%) included a table of the numbers at risk over time, 10 (20%) depicted censoring in some way, either within a table beneath the plot or as ticks on the lines, and 5 (10%) depicted uncertainty using some form of CI.

The objectives of this work are first, to identify alternatives in relation to the above issues, and second, to understand which (if any) alternatives offer improvements to standard practice.

## METHODS
Resulting directly from the objectives, the two activities undertaken were:

1. To propose some improvements in KM plots.
2. To survey stakeholders in order to understand which are preferred.

### Graph development
The constraint on activity (1) was that any proposals should still principally contain a figure showing the KM estimate over time and should not be based on a different visual description of survival data (such as those in Refs. 7 and 8).

A number of proposals were conceptualised, created and triaged. These were taken forward on the basis of being reasonably different to one another and favoured by at least one of the authors. This resulted in six proposals to take to survey, including four alternative means of representing the numbers at risk and two means of representing uncertainty.

### Sources of data and randomisation
With the aim of covering a range of scenarios, we created the proposals for three published, phase III randomised trials:

1. *RT01*: a two-arm trial in prostate cancer which showed a clear difference in biochemical progression-free survival.[4]
2. *ICON7*: a two-arm trial in ovarian cancer with crossing survival curves.[9]
3. *LY09*: a three-arm trial in Hodgkin's lymphoma with limited differences between the arms.[10]

Participants were invited to take a short survey of 13 questions relating to these proposals.

To avoid the repetition and burden of answering all questions for each of the three trials, participants were randomly assigned to see graphs for just one of the trials, using simple randomisation in a 1:1:1 ratio (via a JavaScript tool invoked when a participant clicked the link to take the survey). The purpose of this randomisation was not to compare the randomised groups (as in a randomised trial) but to elicit opinions averaged over these three scenarios.

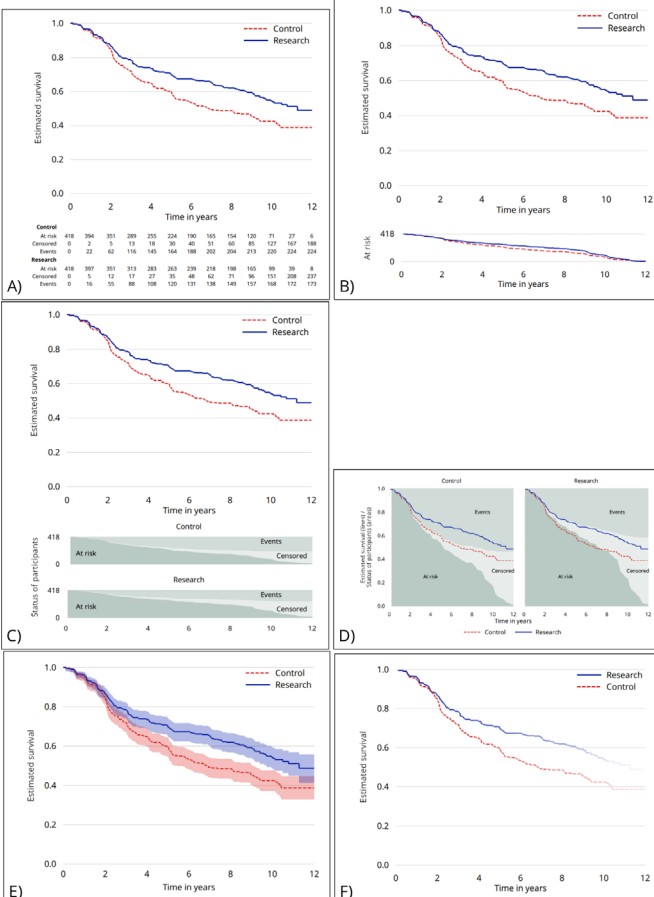

**Figure 2** Proposed graphs using the RT01 data. (A) An extended table showing the status of participants over time. (B) A plot of the number of participants at risk over time by arm. (C) A plot of the status of participants over time by arm, beneath the Kaplan–Meier plot. (D) Two plots of the status of participants over time, one for each arm, behind the Kaplan–Meier plot. (E) CIs presented around the Kaplan–Meier estimate. (F) Fading of the Kaplan–Meier lines as information reduces.

### Survey overview

The survey asked for participants' opinions and preferences regarding the six alternatives as compared with a reference that we regard as a reasonably 'standard' KM plot (similar to figure 1; note that what constitutes standard is subject to opinion). The proposals are shown in figure 2 (for the RT01 trial data, with standard based on authors' consensus). Larger versions of each graph are given in the online supplementary file (online supplementary figures 1–18) for all three trials.

In order to understand which of these proposals were preferred by stakeholders, we conducted a survey using Online Surveys (formerly BOS).[11]

### Taking the survey

Participants were shown each proposed graph and asked to score it on a five-point ordinal scale, against the reference graph (without the proposed alteration, similar to that in figure 1), with the reference and proposal options visible side by side. The options were 'Less useful',

'Equal/no preference', 'A bit more useful', 'Somewhat more useful' and 'Much more useful'. Participants were next asked to rank (in order) up to three preferred proposals and to not rank any proposal that they disliked. This ranking was done separately, once for the proposals addressing the numbers at risk and once for those addressing uncertainty.

After answering each of the above questions, participants had an opportunity to provide free-text comments, and a further opportunity to provide general comments on the survey. This gave a chance to explain their ratings of graphs. All of the free-text comments were read and categorised by the authors, with participants' comments assigned completely at random to one of BCO, CIJ, MJS, TPM or WJC. These comments were categorised in two ways. First, many of the comments were categorised as being to criticise, praise or suggest improvements to one of the proposals (most proposal-specific comments fell into one of these categories). Second, we categorised further comments (not proposal specific) according to the comments made.

### Baseline information collected

As well as opinions, we collected some participant characteristics. For descriptive purposes, we collected the country in which participants are primarily based, the date on which the survey was taken, and the years of experience (1) 'reading and interpreting', and (2) 'creating' KM graphs. To explore whether opinions varied according to two specific characteristics, we also asked what participants identified as their principal professional background and whether or not they currently act as a journal editor. We regard the latter as important because journals often specify styles for KM plots (either in instructions to authors or during typesetting) and so editors may exert disproportionate influence over what appears in the literature.

### Recruiting participants

We recruited participants by publicising the survey through many channels: emails to colleagues and collaborators, Twitter, email lists including AllStat and the ISCB (International Society for Clinical Biostatistics) list, clinical collaborators of the MRC Clinical Trials Unit at UCL, the UK Hubs for Trials Methodology Research (note that this list is non-exhaustive). As the survey ran, we noted the high proportion of participants whose primary role was statistician, and so targeted clinicians and systematic reviewers more purposefully.

### Analysis of results

Analysis of the data is descriptive, generally depicting the frequency of specific responses in graphs. The data on which the analysis is based are provided in the online supplementary file for readers to explore themselves, minus the date of survey, free-text comments and participant country.

### Note on 'sampling'

The survey did not have any formal sampling mechanism (or well-defined units of the population, or its size) and

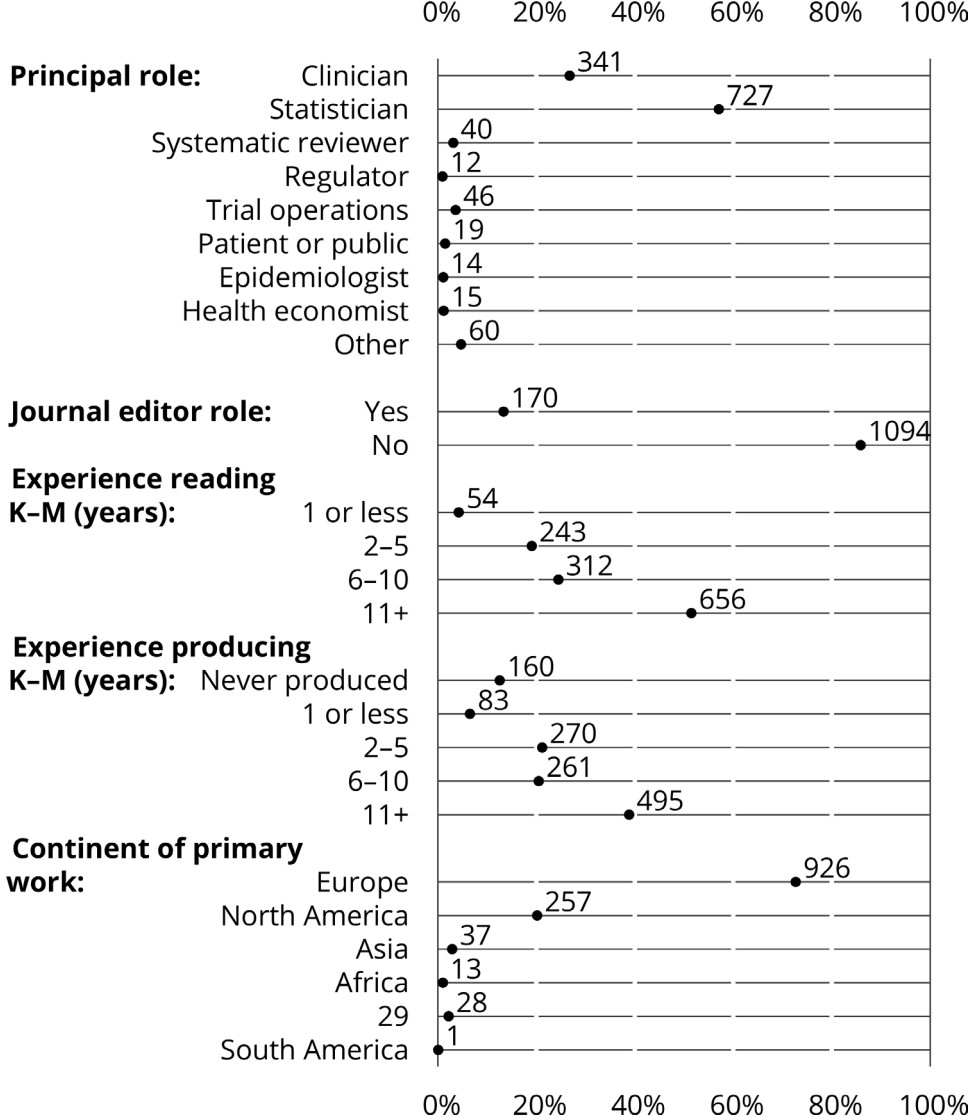

**Figure 3** Descriptive characteristics of participants (n=1274) as a dot chart (% on horizontal axis; frequencies labelled directly). K-M, Kaplan–Meier.

is a convenience sample. We targeted those that we view as users and/or creators of KM plots and who we could reach, for example, registered clinical trials units in the UK, journal editors and systematic reviewers.

### Data availability

De-identified data containing individual responses to the survey will be made openly available on publication. De-identification necessarily required removal of some of the descriptive variables, including free-text comments (some comments made participants identifiable) and country (continent is retained).

### RESULTS

One thousand, two hundred and seventy-four participants completed the survey between 26 April 2017 and 7 July 2017.

Figure 3 gives descriptive information about the participants: self-described primary role/training, country in

which they primarily work, whether they act as a journal editor, and experience (1) reading/interpreting and (2) producing KM plots. Online supplementary figure 19 is a plot of experience *reading/interpreting vs producing* KM plots.

The most represented roles were statistician (727; 57%) and clinician (341; 27%). Several other groups were well represented (see figure 3), but the results will be dominated by the groups identifying themselves as statistician or clinician. One hundred and seventy (14%) respondents identified themselves as journal editors. Participants were based primarily in the UK and USA but 36% were based in other countries, representing all populated continents (see figure 3).

Participants' opinions on the proposed alterations to KM plots are given in figure 4. The upper row of figure 4A contains the proposals for presenting how the number at risk changes over time, the lower row of figure 4A for those depicting uncertainty. On the upper

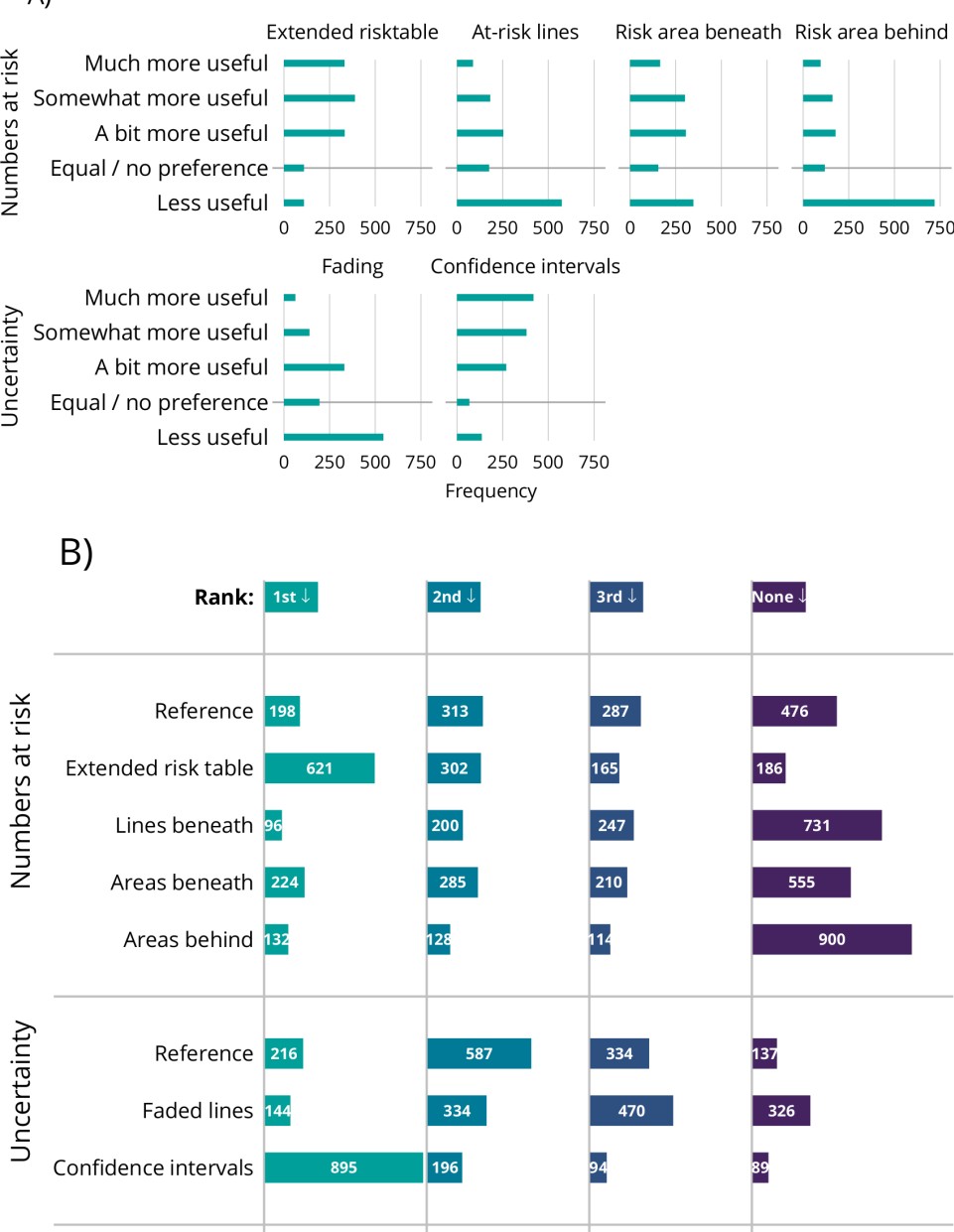

**Figure 4** (A) Opinion of alteration versus 'standard' Kaplan–Meier plot. Upper row is for alterations in presenting numbers at risk; lower row is for alterations in depicting uncertainty. (B) Participants' overall preferences for presenting numbers at risk (upper part) and depicting uncertainty (lower part).

row, the extended risk table garnered the most positive opinions, with 1054 (83%) participants giving a positive response. Some (110 participants; 9%) found this extra information 'less useful'. Using a line graph to depict the numbers at risk was not popular, with 574 (45%) finding this less useful than the usual table depicting numbers at risk. The graph of areas to replace the extended risk table divided opinion: while 347 (27%) found it less useful than a standard plot, 772 (61%) found it 'a bit', 'somewhat' or 'much more' useful. The same chart with the areas superimposed behind the KM estimate was much less popular, with 720 (57%) finding it less useful than the usual plot (at this first exposure). The lower row shows ambivalence about the idea of faded lines: 545 (43%) found this less

useful than a standard presentation, 195 (15%) had no preference and 534 (42%) found it more useful.

These results were broadly similar across the three trials, both for statisticians and clinicians, and for editors and non-editors of journals. Figures similar to the upper panel of figure 4, broken down by these groups, can be found in the online supplementary file.

Figure 4B gives participants' overall rankings for the proposals, separately for those addressing numbers at risk and uncertainty. Green bars depict the number of participants who ranked this graph as their first choice; orange as second, red as third and grey not ranked (for proposals depicting uncertainty, although there were only three options, participants did not have to answer

for all choices if, eg, they found only one option to be acceptable). These results give the same message as those presented in figure 4A. For presenting numbers at risk, an extended risk table is the clear favourite; for depicting uncertainty, the use of CIs was the first choice for over half the participants.

An idea of the nature of free-text responses is provided in online supplementary figure 20, which summarises whether graph-specific comments were criticism, praise or suggestions (left) and gives the broad types of comment (right).

## DISCUSSION

We have proposed several alterations to standard KM plots, specifically for the context of showing within-arm survival in randomised trials. The proposals were around two key aspects depicting: (1) the numbers at risk over time and (2) uncertainty.

We then surveyed users of KM plots for their views on our proposals. Several garnered more positive opinions than the reference plot, and two came out as the overall favourites, although opinions were far from unanimous.

We do not make explicit recommendations here about which alterations should be used but encourage producers of KM plots and those who influence them (journal editors and regulators) to consider their practice in light of these results. In particular, the plots including an extended table of numbers and CIs seemed to be favoured by most participants. These can be used in combination without any clash, and we include an example with both aspects in figure 5, again using the RT01 data.

There is clear recognition that graphical representations of time-to-event data could be improved. Many free-text responses noted context. KM plots are used by: trial

designers looking for previous information on a related group of patients; data monitoring committees viewing interim data; meta-analysts to extract data and clinicians looking to understand and communicate risks to their patients. There is no one size that fits all settings and producers of KM plots need to make judicious choices according to their context.

The two proposals involving area graphs to depict the number at risk require some thought to understand and are not instantly readable; a graph that requires little 'learning to read' is perhaps desirable. These two proposals were broadly unpopular in the survey: many commented that this depiction was confusing, but a minority who liked them said it took time to reach that conclusion. Prior to the survey, the authors had expected the KM curve superimposed on the area depicting numbers at risk to be more popular than they were. The results of the survey show the desirability of a graph that requires little 'learning to read' and also the importance of a large stakeholder survey to elicit representative preferences.

Depicting the numbers at risk using line charts below the KM plot was also reasonably unpopular. Free-text comments suggested three main reasons: (1) participants wanted specific numbers in preference to a general pattern; (2) the line looks similar to the line of the KM estimate, leading to potential confusion; and (3) as we created and presented this option, the plot region for the numbers in follow-up used 1/3 the area of the plot region for the KM estimate, which for some participants was inadequate—a poor choice on our part. This proportion would need to be reconsidered by anyone looking to use the approach.

For depicting uncertainty, fading the KM estimates was unpopular. There were two principal reasons for this. First, when printed, the fading could be confused for a printing error, rather than an intended effect. Second, it is not clear how to define the level of decreasing intensity that accurately reflects the readers' perception of increasing uncertainty. A minor comment from some clinicians was the desire to be able to accurately read the estimate at a very late time point (note that the premise for use of fading was in part to prevent this where uncertainty is extremely high).

Further thought is required on visualising survival data, and new proposals would ideally be accompanied by studies on stakeholders' opinions. We constrained this project to KM plots with two or more groups. However, in a randomised trial, we are interested in comparing arms and so want to visualise some estimate of the difference. Such visualisations may be a fruitful future direction.

Interestingly, Paul Meier himself is said to have spoken with bemusement about people plotting KM estimates over time and was not convinced he actually liked it (the authors thank Chris Barker, a former student of Meier, for this personal communication).

As noted in the Methods section, the trial datasets we used do not represent any true distribution of scenarios

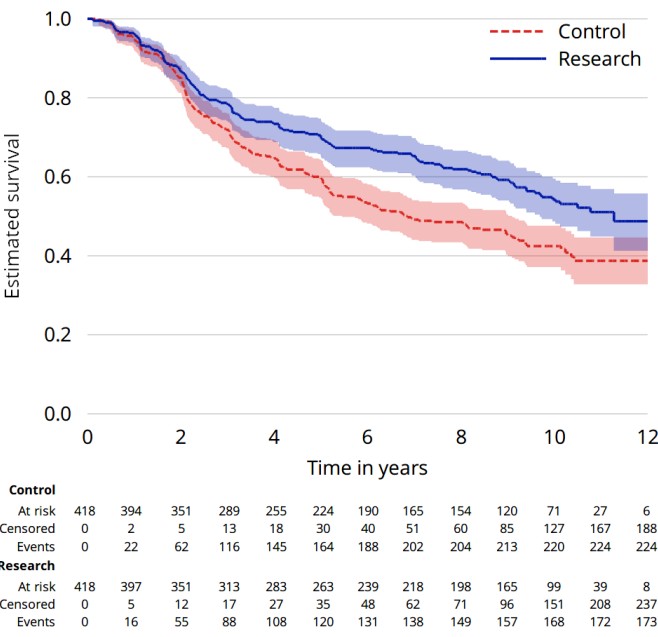

| Control | | | | | | | | | | | | | |
|---|---|---|---|---|---|---|---|---|---|---|---|---|---|
| At risk | 418 | 394 | 351 | 289 | 255 | 224 | 190 | 165 | 154 | 120 | 71 | 27 | 6 |
| Censored | 0 | 2 | 5 | 13 | 18 | 30 | 40 | 51 | 60 | 85 | 127 | 167 | 188 |
| Events | 0 | 22 | 62 | 116 | 145 | 164 | 188 | 202 | 204 | 213 | 220 | 224 | 224 |
| **Research** | | | | | | | | | | | | | |
| At risk | 418 | 397 | 351 | 313 | 283 | 263 | 239 | 218 | 198 | 165 | 99 | 39 | 8 |
| Censored | 0 | 5 | 12 | 17 | 27 | 35 | 48 | 62 | 71 | 96 | 151 | 208 | 237 |
| Events | 0 | 16 | 55 | 88 | 108 | 120 | 131 | 138 | 149 | 157 | 168 | 172 | 173 |

**Figure 5** The two most popular elements combined: CIs and extended at-risk table.

occurring in clinical trials. Rather, they represent a small variety of situations which can occur in randomised trials; for any change to KM plots to be worthwhile, the impact of features such as non-inferiority, non-proportional hazards, more than two arms and different allocation ratios should be assessed. Having said this, if a plot works well for two-arm trials but not three-arm trials, it may of course be used in that context.

We hope that this work will provoke those creating KM plots to think carefully about how they can best convey the information, and that journal editors will consider their policies for rendering KM plots. We will continue to consider alternatives and evaluate these in the future.

## Patient and public involvement
There was no formal patient and public involvement in the development of the KM proposals but patients were an important group of participants who were actively targeted by our survey. Alongside the researchers, 19 patients participated. Our objective was to improve researchers' understanding of KM plots. The survey itself was an attempt to involve such researchers by asking for their views on our proposals.

## Dissemination declaration
We will email results to all survey participants who stated that they wished to be updated. This will be done to coincide with publication of the manuscript.

**Acknowledgements** The authors are grateful to the 1274 participants who provided their opinions, particularly those who pointed us to articles containing their own thoughts on Kaplan–Meier; to the ICON7, LY09 and RE01 trial teams and participants for the use of these data.

**Contributors** The initial idea for this project came from TPM and MRS. All authors were involved in the initial planning, design and running of the study. TPM and CIJ checked and cleaned the data and TPM analysed the results. All authors commented critically on and approved the manuscript.

**Funding** This work was supported by the Medical Research Council (grant numbers MC_UU_12023/21, MC_UU_12023/29 and MC_UU_12023/24)—TPM, WC, PP, BC-O, MRS.

**Competing interests** None declared.

**Patient consent for publication** Not required.

**Ethics approval** No ethical approval was required (or obtained), assessed using the online HRA decision tool. This was a survey of opinions on a non-sensitive subject, collecting no biological samples or data that might be identifiable (unless participants identified themselves in a free-text comment or chose to provide contact details to hear about the results of the survey).

**Provenance and peer review** Not commissioned; externally peer reviewed.

**Data availability statement** All shareable data relevant to the study are available on the UCL Research Data Repository.

**ORCID iDs**
Tim P Morris http://orcid.org/0000-0001-5850-3610
Christopher I Jarvis http://orcid.org/0000-0002-0812-2446
William Cragg https://orcid.org/0000-0002-1274-8521
Patrick P J Phillips https://orcid.org/0000-0002-6336-7024
Babak Choodari-Oskooei https://orcid.org/0000-0001-7679-5899
Matthew R Sydes http://orcid.org/0000-0002-9323-1371

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
