## [Reviewer comments · BMJ Open]

ARTICLE DETAILS

TITLE (PROVISIONAL)	Proposals on Kaplan–Meier plots in medical research and a survey of stakeholder views: KMunicate
AUTHORS	Morris, Tim; Jarvis, Christopher; Cragg, William; Phillips, Patrick; Choodari-Oskooei, Babak; Sydes, Matthew

VERSION 1 – REVIEW

REVIEWER	Keri Calkins Postdoctoral Fellow Johns Hopkins University Department of Epidemiology Bloomberg School of Public Health Baltimore, MD, USA
REVIEW RETURNED	23-Apr-2019

GENERAL COMMENTS	You provided a good summary of the survey and results. I agree that the confidence intervals and extended at-risk table are probably the most readable to a wide scientific audience. The conclusion in your abstract seems more definitive than the one in the paper in terms of using this figure as a new standard. I would perhaps change this phrasing to mirror your stated purpose of provoking thought of those creating KM plots and incorporating stakeholders opinions.
--

REVIEWER	Jesus Orbe University of the Basque Country Spain
REVIEW RETURNED	02-May-2019

GENERAL COMMENTS	Comments and Suggestions for the Authors Summary Contents The objective of the paper is study the way to present time-to-event data, such as survival, using Kaplan–Meier plots. The authors compare the standard way to present time-to-event data or survival with Kaplan–Meier plots against other different Kaplan-Meier plots. The authors focus their interest on two aspects. On the one hand, the state of the patients over time, i.e. whether they are at risk, censored or the event has occurred and on the other hand, in the uncertainty of the estimation of the survival function over time. They compare standard Kaplan-Meier plot and six alternative Kaplan-Meier plots. The first alternative adds the usual table of
--

	number at risk, censored and events below the standard plot. The second alternative replaces the usual table by a line graph of the numbers at risk over time. The third one presents the same information as the first one but using a graphical form. The fourth alternative uses the graphs of the third alternative but behind the Kaplan-Meier plot. Confidence intervals are plotted around the Kaplan-Meier in the fifth alternative and, in the last one, the Kaplan-Meier curves fade in proportion to the cumulative number of censored individuals. In summary, the first four alternatives presented show practically the same information. The fifth one focuses more on the uncertainty of the estimate over time and the sixth one could be confused for a printing error. The authors use a survey asking for participants' opinions and preferences regarding the six alternatives as compared with the reference that they regard as a standard Kaplan-Meier plot. The rest of the work is a descriptive study of the obtained participants' opinions. In the Kaplan-Meier plot the estimation of some probabilities is represented. Therefore, for a good interpretation of an estimation, the estimated values are necessary, as well as an indicator of their reliability or uncertainty through, for example, the confidence intervals. A good example would be Figure 5. Specic Corrections and Suggestions The text needs more careful reading to avoid some typos. The results section is not well understood until the supplementary information is read. In order to improve the reading of the work it might be helpful if the subfigures included in figure 2 would have titles so that they could be referenced in the text. Otherwise the reading becomes quite complicated. Figure 3 could be summarized with a table of percentages. The reference to the article by Kaplan, E. L. and Meier, P. (1958) proposing this estimation method is missing.
--	--

REVIEWER	Moritz Berger University of Bonn
REVIEW RETURNED	05-Jun-2019

GENERAL COMMENTS	Comments on: Proposals on Kaplan-Meier plots in medical research and a survey of stakeholder views: KMunicate, submitted to BMJ Open The paper presents the results of a survey evaluating several modifications of a standard KaplanMeier plot to improve the informative value of the plots. The proposals on adding information about the "number at risk" and the "uncertainty" are useful and the results of the survey seem helpful for practitioners. However, there are several weaknesses regarding the presentation of the results and a lack of clarity in some parts. Major Remarks  1. The descriptive characteristics in Figure 3 and the results in Figure 4 are presented in an inadequate/incomprehensible way. Using barplots that show the absolute/relative frequencies on the
--

	y-axis and the categories on the x-axis would be a much better choice. Stacked or side-by-side barplots would be the best choice for the upper panel of Figure 4. 2. The six different proposals are shown in Figure 2, but they are neither described in sub captions nor described in the text (this could be added to the Section "Graph development"). Moreover, it is hard to match the six plots in Figure 2 to the results in Figure 4. I recommend using the same headers. 3. The survey was conducted using three randomized trials. However, all the Kaplan-Meier plots shown in the article are based on the first trial (RT01). It would be very interesting to also see the (modified) plots for the other two trials. 4. The mentioned target sample size is irritating, because (i) the realized sample size is much higher, and (ii) the authors say that the choice was pragmatic. In my opinion, this part should be omitted. The statistical analysis is purely descriptive and no statistical tests (e.g. to compare the ordinal scales) was conducted, which requires a sample size consideration. 5. The last step of the survey ('Participants were next asked to rank (in order) up to three of their preferred graphs) is not clear to me. These results are not shown in Figure 4, are they? Did you treat the 'number at risk' and the 'uncertainty' proposals separately or together when evaluating the results? 6. Page 9, Line 13f: The figures that are described here are not included in the supplementary file! Minor Remarks a. Page 6, Line 9: Why did you review the articles from 2013? This is maybe not the current status. b. Page 7, Line 21: Please give the levels of the ordinal scale in the text. c. Page 8, Line 15: Analysis of the data 'was done'. d. Page 8, Line 57: What is meant by 'positive response'? e. Page 9, Line 27: which figure? f. Page 8, Line 60, Page 9, Line 45: Typos in '1nding' and 'in2uence'.
--	--

VERSION 1 – AUTHOR RESPONSE

Reviewer 1: Keri Calkins, Johns Hopkins University, MD, USA

'I would perhaps change this phrasing to mirror your stated purpose of provoking thought of those creating KM plots and incorporating stakeholders' opinions.'

We agree: the abstract was rather more emphatic than the stated purpose. We have now toned this down by removing the final sentence.

Reviewer 2: Jesus Orbe, University of the Basque Country, Spain

'The text needs more careful reading to avoid some typos.'

Thank you for noting this. Unfortunately this occurred due to conversion between Latex and Word after initially submitting a Latex-generated pdf. We have checked and hope we got them all this time!

'The results section is not well understood until the supplementary information is read. In order to improve the reading of the work it might be helpful if the subfigures included in figure 2 would have titles so that they could be referenced in the text. Otherwise the reading becomes quite complicated.'

Thank you. This is a really helpful suggestion to improve clarity. We have now labelled the sub-figures with letters, and the descriptions in the text use these labels.

'Figure 3 could be summarized with a table of percentages.'

This article is about graphical representation of data and we opted to use a Cleveland-style dot chart to summarise the information. We have amended the figure to make it cleaner – the previous water-marks added by the submission system certainly added visual clutter!

'The reference to the article by Kaplan, E. L. and Meier, P. (1958) proposing this estimation method is missing.'

Thank you for picking this up. Kaplan and Meier themselves never considered the plot – one of Meier's students got in touch to say that Meier himself was 'bemused' by the plot! – but it seems fair to include the reference and we have now done so, along with this anecdote.

Reviewer 3: Moritz Berger, University of Bonn

'1. The descriptive characteristics in Figure 3 and the results in Figure 4 are presented in an inadequate/incomprehensible way. Using barplots that show the absolute/relative frequencies on the y-axis and the categories on the x-axis would be a much better choice. Stacked or side-by-side barplots would be the best choice for the upper panel of Figure 4.'

We originally started with a stacked bar chart for figure 4a) as the reviewer describes and found it rather more incomprehensible. A frequent criticism of stacking bars is that it is very difficult to read the relative size of ranks as one moves across categories. Further, people are better at perceiving horizontal than vertical motion, which is why histograms should use vertical bars but bar charts very frequently use horizontal bars, and why we have represented participants' opinions on the horizontal axis. We have now edited figure 4b) because the stacking there made the results rather hard to compare across ranks.

'2. The six different proposals are shown in Figure 2, but they are neither described in sub captions nor described in the text (this could be added to the Section "Graph development"). Moreover, it is

hard to match the six plots in Figure 2 to the results in Figure 4. I recommend using the same headers.'

Thank you. This is an important comment, also made by Reviewer 2. We have now labelled each sub-plot with a letter so it is clear which plot is which. This will help to avoid confusion

'3. The survey was conducted using three randomized trials. However, all the Kaplan-Meier plots shown in the article are based on the first trial (RT01). It would be very interesting to also see the (modified) plots for the other two trials.'

These are now included in supplementary materials, which we should have done originally.

'4. The mentioned target sample size is irritating, because (i) the realized sample size is much higher, and (ii) the authors say that the choice was pragmatic. In my opinion, this part should be omitted. The statistical analysis is purely descriptive and no statistical tests (e.g. to compare the ordinal scales) was conducted, which requires a sample size consideration.'

We accept this comment and this has been removed.

'5. The last step of the survey ('Participants were next asked to rank (in order) up to three of their preferred graphs) is not clear to me. These results are not shown in Figure 4, are they? Did you treat the 'number at risk' and the 'uncertainty' proposals separately or together when evaluating the results?'

These are the results shown in figure 4. The two 'groups' of proposals were treated separately (hence the separating line in the figure), which we have tried to clarify further both in the text and in the results. Participants would rank up to three of the proposals about the number-at-risk, then up to three of the proposals about uncertainty (participants were asked to provide no ranking for proposals they disliked, so that we could view 'any ranking' as positive).

'6. Page 9, Line 13f: The figures that are described here are not included in the supplementary file!' Apologies. We submitted files similar to figure 4 broken down by subgroups, which perhaps did not appear for reviewers. However, rather than including these analyses in the manuscript, the cleaned data from the survey will instead be made openly available so that readers can, if they wish, conduct their own analyses. The cleaned data file was uploaded with this submission, but the manuscript system bounced the submission because the data file was not in pdf format. We have since understood that the journal will not include data as a supplementary file and so will upload to figshare on publication.

'a. Page 6, Line 9: Why did you review the articles from 2013? This is maybe not the current status.' One of the authors (BCO) was involved in a separate review using the 2013 papers (doi:10.1186/s13063-019-3251-5), also published in 2019. He used plots taken from those articles for a snapshot of what people were doing in 2013, which we regard as more informative than no snapshot. We are not aware of any other recommendations or major change in the journals' editorial policies since 2013.

'b. Page 7, Line 21: Please give the levels of the ordinal scale in the text.' This has been done.

'c. Page 8, Line 15: Analysis of the data 'was done'.' Ok.

'd. Page 8, Line 57: What is meant by 'positive response'?' Anything better than 'equal/no preference' (described in the 'Taking the survey' section).

'e. Page 9, Line 27: which figure?'

Apologies. Now changed to 'supplementary figure 1'

'f. Page 8, Line 60, Page 9, Line 45: Typos in '1nding' and 'in2uence'.'

Apologies. This happened during conversion from Latex to Word (due to 'fi' and 'fl' ligatures respectively) and has been corrected (we hope everywhere).

VERSION 2 – REVIEW

REVIEWER	Moritz Berger University of Bonn
REVIEW RETURNED	09-Aug-2019

GENERAL COMMENTS	I have no further comments.
-----------------------------